# Assessment of Human Gingival Fibroblast Proliferation after Laser Stimulation In Vitro Using Different Laser Types and Wavelengths (1064, 980, 635, 450, and 405 nm)—Preliminary Report

**DOI:** 10.3390/jpm11020098

**Published:** 2021-02-04

**Authors:** Barbara Sterczała, Kinga Grzech-Leśniak, Olga Michel, Witold Trzeciakowski, Marzena Dominiak, Kamil Jurczyszyn

**Affiliations:** 1Dental Surgery Department, Wroclaw Medical University, 50-425 Wroclaw, Poland; marzena.dominiak@umed.wroc.pl (M.D.); kjurczysz@interia.pl (K.J.); 2Laser Laboratory at Dental Surgery Department, Wroclaw Medical University, 50-425 Wroclaw, Poland; kgl@periocare.pl; 3Department of Periodontics, School of Dentistry, Virginia Commonwealth University, VCU, Richmond, VA 23298, USA; 4Department of Molecular and Cell Biology, Wroclaw Medical University, 50-556 Wroclaw, Poland; WF-26@umed.wroc.pl; 5Institute of High Pressure Physics, Polish Academy of Sciences, 01-142 Warsaw, Poland; wt@unipress.waw.pl

**Keywords:** cell proliferation, human gingival fibroblasts, photobiomodulation, PBM, soft tissue regeneration

## Abstract

Purpose: to assess the effect of photobiomodulation (PBM) on human gingival fibroblast proliferation. Methods: The study was conducted using the primary cell cultures of human fibroblasts collected from systemically healthy donors. Three different laser types, Nd:YAG (1064 nm), infrared diode laser (980 nm), and prototype led laser emitting 405, 450, and 635 nm were used to irradiate the fibroblasts. Due to the patented structure of that laser, it was possible to irradiate fibroblasts with a beam combining two or three wavelengths. The energy density was 3 J/cm^2^, 25 J/cm^2^, 64 J/cm^2^. The viability and proliferation of cells were determined using the (Thiazolyl Blue Tetrazolium Blue) (MTT) test conducted 24, 48, and 72 h after laser irradiation. Results: The highest percentage of mitochondrial activity (MA = 122.1%) was observed in the group irradiated with the 635 nm laser, with an energy density of 64 J/cm^2^ after 48 h. The lowest percentage of MA (94.0%) was observed in the group simultaneously irradiated with three wavelengths (405 + 450 + 635 nm). The use of the 405 nm laser at 25 J/cm^2^ gave similar results to the 635 nm laser. Conclusions: The application of the 635 nm and 405 nm irradiation caused a statistically significant increase in the proliferation of gingival fibroblasts.

## 1. Introduction

Nowadays, laser has become an indispensable tool in the dental office [1,2,3]. It is used as a treatment device that has replaced surgical instruments such as the scalpel; it improves the patient’s postoperative comfort by minimizing pain and swelling [4]. Depending on the type, the laser may be used in the area of soft and hard tissues. It is used, for instance, in orthodontics, in the treatment of temporomandibular joint dysfunctions and implantological rehabilitation [5,6,7,8,9]. Moreover, lasers have been demonstrated to support the treatment of snoring and sleep apnea, as well as periodontal diseases, e.g., peri-implantitis [10,11,12,13,14,15,16,17]. The effects of laser photobiomodulation (PBM) in muco-gingival surgery are relatively well established [18,19,20].

The mechanism of action of laser PBM at the cellular level is based on the simultaneous activation of many reactions. The structure and function of biological membranes are modified, endorphins are released, and an increase in the activity of the immune system of cells and enzymes is observed [21]. A signaling cascade is triggered by the electron transfer between five protein complexes found in the mitochondrial membrane. The above mentioned membrane proteins include NADH dehydrogenase, succinate dehydrogenase, cytochrome c reductase, cytochrome c oxidase, and adenosine triphosphate (ATP) synthetase, as well as free molecules like ubiquinone and cytochrome c. Metabolic processes are activated in the mitochondria that stimulate chromophores in the mitochondrial respiratory chain and induce an increase in the production of ATP [22,23,24]. The intensification of cellular respiration contributes to the growth, migration, differentiation, and proliferation of cells within the wound, including fibroblasts. The bioactive mediators released by fibroblasts, e.g., growth factors, affect the metabolism of cells by synthesizing components of the extracellular matrix such as collagen type I [25,26,27]. In addition to being important for wound healing or tissue regeneration, PBM influences angiogenesis by generating vascular endothelial growth factor (VEGF) [28,29]. Changes at the cellular and biochemical levels can evoke therapeutic effects due to the stimulation of wound healing, anti-inflammatory and analgesic action, as well as prevention of swelling and tissue necrosis [30].

The improved knowledge of tissue biomimetics has introduced significant changes in periodontological plastic surgery, which contributed to the development of new tools in the form of primary cell culture of fibroblasts [31]. This method involves obtaining autogenous tissue in quantities determined by its shortage. Isolated and cultured gingival fibroblasts can be implanted on various types of matrices, both xenogenic (collagen 3D) and allogenic (fascia lata) ones [32]. In the form of a hydrated cell matrix, they are implanted in the recipient site, e.g., the augmentation of the keratinized gingival performed on a thin biotype before planned orthodontic treatment, implant, or prosthetic rehabilitation [33]. This method undoubtedly makes it possible to reduce the invasiveness of the procedure resulting from the minimal recipient site (2 mm^2^) and gives quick final results thanks to short breeding time which is up to 10 days. In addition, the site of augmentation does not lead to excessive tension or thickening of the mucous membrane and the creation of scarring features, which is important in the aesthetic aspect of treatment [34]. This method may be a clinically satisfactory alternative to autogenous connective tissue transplantation, while maintaining the indication for its use. Importantly, this method makes it possible to obtain the optimal amount of fibroblasts. In their histological studies, Dominiak et al. (2010) proved that the optimal amount of autogenous fibroblasts implanted on the collagen matrix accelerates the wound healing process and affects the generation of angiogenesis and the elimination of features of scarring [35]. The PBM mechanism indicates that this is an appropriate technique to obtain more fibroblasts, and indirectly, growth factors for soft tissue regeneration. Therefore, research is being conducted to optimize irradiation parameters in specific clinical procedures. Most reports represent the use of radiation for biostimulation in the range of low-power light which should not increase the temperature by more than 5 °C. An increase in the temperature may lead to significant changes in the structure of the cell membrane, such as the minimized activity of the sodium-potassium pump and reduced flow of calcium ions, which in turn leads to a decrease in the amount of cyclic ATP. Reduced oxidative phosphorylation and limited DNA and RNA synthesis result in the reduction of cell proliferation and functioning of the immune system.

The aim of this in vitro study was to compare the effect of different wavelengths: 405 nm, 450 nm, and 635 nm as well as their combinations—980 nm and 1064 nm on the proliferation of human fibroblasts.

## 2. Materials and Methods

### 2.1. Cell Culture

This study was approved by the Ethics Committee of Wroclaw Medical University, Poland (No KB-434/2017).

The study was conducted using the primary cell cultures of human fibroblasts collected from systemically healthy donors, from the hard palate. Tissue was collected from an area of 1–2 mm^2^ and transported to the laboratory in the nutrient medium Dulbecco’s Modified Eagle Medium (DMEM, Sigma-Aldrich, Poznan, Poland) with the addition of 10% fetal calf serum, penicillin (100 Ul/mL), streptomycin (0.1 mg/mL), and amphotericin B (0.1 mg/mL). Next, fibroblasts were isolated and cultured according to the procedure developed by Polish scientists M. Dominiak and J. Saczko, Patent No. 381204 (Saczko, 2008), based on the mechanical isolation, which affects cellular biomechanics [34]. The culture was carried out in a conventional DMEM culture medium in an incubator at 37 °C in a 5% CO_2_ atmosphere. The culture medium was changed twice a week. The cells reached a full monolayer after 5–7 days. For the purposes of laser irradiation, fibroblasts were subjected to trypsinisation (0.25% trypsin-EDTA, Sigma-Aldrich, Poznan, Poland) and seeded into 96-well plates for further examination.

### 2.2. Laser Irradiation

Three lasers were used to irradiate fibroblasts. One of them was the Nd:YAG laser (LightWalker, Fotona, Slovenia) with a flat-top handpiece (Genova, LightWalker, Fotona, Slovenia). This handpiece produces a spot with a diameter of 10 mm (which is exactly the diameter of the wells in the culture plates) and with a homogeneous beam profile. The other two lasers were a diode laser of 980 nm (Smart M, Lasotronix, Poland, Piaseczno, Poland), and a diode laser constructed by Trzeciakowski et al. from the Polish Academy of Sciences (Patent No.: 9,223,123). Eight laser diode beams are coupled into a multi-mode fiber (with a 100–400 μm core) using a reflector in the form of a regular pyramid (Ivonyak et al., 2014). The optimization of the optical system enables the coupling of 60–90% of light into the fiber. In the laser developed for the project, we used three wavelengths emitted by 8 diodes:3 diodes emitting at 405 nm (maximum output power of 0.70 W),2 diodes emitting at 450 nm (maximum output power of 1.0 W), 3 diodes emitting at 635 nm (maximum output power of 1.0 W).

Since all laser diodes were coupled into the same fiber, it was possible to irradiate fibroblasts with a beam combining one, two, or three wavelengths. Exposure was carried out using the headset at right angle to the samples and at a distance of 5 mm from the cells. The parameters used in the study are presented in Table 1. In case of both commercial lasers (Light Walker Fotona and Smart M, Lasotronix), the output power was determined on the basis of energy density, so we set power settings of the prototype laser to values similar to those of the commercial laser parameters. To achieve the same energy density (ED), the time of irradiation was varied. In case of pulsed mode in all lasers, the frequency was 10 Hz, and the duty cycle was 50%. In case of two or three wavelengths in one irradiation, the power was equally distributed between each wavelength.

Before the irradiation, the cells were seeded on the black 96-well plates, at ~5 × 10^3^ cells per well. The distribution of cells in the wells also depended on power to avoid accidental exposure of cells from an adjacent well. In the case of an energy density of 3 J/cm^2^, the cells were located every third well, in case of energy density of 25 J/cm^2^—every six wells, and in case of energy density of 64 J/cm^2^—diagonally on a plate. For each wavelength and energy density, three independent exposures were performed for three independent samples. The cells were incubated for 24 h under standard conditions (as above) and then subjected to laser irradiation (PBM).

### 2.3. Cell Proliferation MTT Test

The viability and proliferation of cells were determined by MTT test (Sigma-Aldrich) at 24, 48, and 72 h after laser irradiation. The method allows for determining cell survival by measuring the mitochondrial activity of the tested cells. After removal of the culture medium above cells, 100 μL of MTT solution (0.5 mg/mL) was added to each well (including one well without cells that served as a blank control). Then, the plates were incubated for 2 h in the CO_2_ incubator. As a result of the reaction from the water-soluble MTT salt (tetrazole salt (3-(4,5-dimethylthiazol-2-yl)-2,5-diphenyltetrazolium bromide)) with succinate dehydrogenase, formazan, dark purple, insoluble crystals were formed. Further, crystals were dissolved by the addition of 100 μL of acidic isopropanol to each well. The number of crystals formed was proportional to the amount of mitochondrial succinate dehydrogenase that is active in living cells. The absorbance of the colored solution was measured spectrophotometrically at 570 nm, on a Multiskan™ FC microplate photometer (Thermo Scientific, Alab, Warsaw, Poland). Each tested parameter was performed in triplicate and repeated three times within the independent experiments. 

### 2.4. Statistical Analysis

The statistical analysis was performed using Statistica version 13.3 (Stat Soft, Krakow, Poland). The normal distribution was confirmed using the Shapiro–Wilks test. The presence of statistical differences was determined based on the analysis of variance and the Turkey post hoc test, and the statistical significance level was 0.05. The analysis of variance for multivariate classification was used to evaluate the effect of two controlled factors on the mitochondrial activity of cells (% of MA) (dependent variable Y). The independent variables for each laser included: laser radiation dose (variable X1—surface energy density GE: 0, 3, 25 and 64 J/cm^2^) and the incubation time in the culture fluid (variable X2—three lengths: 24, 48 and 72 h) and their interactions.

## 3. Results

A decrease in the mitochondrial activity (% of MA) below 100% was observed in all groups 24 h after irradiation with the 405 nm laser (3 J/cm^2^, 25 J/cm^2^, 64 J/cm^2^). A statistically significant difference in comparison to the control group was recorded for an energy density of 64 J/cm^2^. Regardless of the energy density, the highest increase in the percentage of MA was observed 48 h after the exposure (MA > 115%); during this period, its statistical differences were shown in relation to the control group. The highest percentage of MA (117.9%) was observed after 48 h at an energy density of 25 J/cm^2^ (without statistically significant differences from other energy densities: 3 J/cm^2^ and 64 J/cm^2^). After 72 h, the % of MA decreased again to a value not statistically different from the control group. The results for MA in this group are shown in Table 2 and Figure 1. The interaction between energy density and incubation time after irradiation for mitochondrial activity is shown in Figure 2a.

In case of the group treated with the 450 nm laser, a statistically significant decrease in the mitochondrial activity was observed 24 h after the exposure in the irradiated subgroup with an energy density of 3 J/cm^2^ (85.6%) and 64 J/cm^2^ (87.3%) compared to the control group. MA values for the 25 J/cm^2^ subgroup did not differ statistically from the control group. After 48 h of exposure, it was observed that the energy density of 3, 25 and 64 J/cm^2^ led to an increase in the mitochondrial activity above 100% but without statistical differences compared to the control group. MA percentage measurements after 72 h did not show a statistically significant difference from the control group. The highest MA value was reached after 72 h in the group exposed to an energy density of 25 J/cm^2^. The results for MA in this group are shown in Table 2 and Figure 3. The interaction between energy density and incubation time after irradiation for mitochondrial activity is shown in Figure 2b.

In the group irradiated with the 635 nm laser, after 24 h there was an increase in the MA for all energy densities (3 J/cm^2^, 25 J/cm^2^, 64 J/cm^2^), but without statistically significant differences in relation to the control group. After 48 h, the mitochondrial activity reached the highest value for an energy density of 64 J/cm^2^ (122.1%). This measurement showed statistical differences from the control group. A decrease in the MA was observed 72 h after the exposure for all energy densities. The results for MA in this group are shown in Table 2 and Figure 4. The interaction between energy density and incubation time after irradiation for mitochondrial activity is shown in Figure 2c.

In the group simultaneously irradiated with two wavelengths (405 + 635 nm), 24 and 48 h after the exposure, the MA dropped below 100% without statistical differences compared to the control group. A slight increase in the mitochondrial activity was observed 72 h after the exposure without statistically significant differences in comparison to the control group. After this period, the highest MA value (102.5%) was also observed in the irradiated group with an energy density of 64 J/cm^2^. The results showing MA values with simultaneous irradiation with two wavelengths (405 + 635 nm) are presented in Table 2 and Figure 5. The interaction between energy density and incubation time after irradiation for mitochondrial activity is shown in Figure 2d.

In the group simultaneously irradiated with two wavelengths (450 + 635 nm), after 24 and 48 h for all energy densities, a decrease in MA below 100% was noted, without a statistical difference compared to the control group. It seems interesting that despite the initial decrease in the MA value, it slightly increased to 103.8% after 72 h in the group irradiated with an energy density of 25 J/cm^2^. However, this difference was not statistically significant. The results regarding the MA value using two wavelengths (450 + 635 nm) are presented in Table 2 and Figure 6. The interaction between energy density and time of incubation after irradiation for mitochondrial activity is shown in Figure 2e.

The simultaneous irradiation with three wavelengths (405 + 450 + 635 nm) resulted in a significant decrease below 100% in MA activity after 24 h. This decrease was observed for all energy densities and was statistically significant in relation to the control group. The largest decrease was observed in the group irradiated with energy density of 25 and 64 J/cm^2^. In both groups, the MA dropped below 50%. After 48 h, there was an increase in mitochondrial activity to the level of the control group when exposed to an energy density of 3 and 64 J/cm^2^. When exposed to an energy density of 25 J/cm^2^, the MA value still showed a statistical difference from the control. After 72 h, the 25 and 64 J/cm^2^ groups showed no change as opposed to the next MA decrease in the 3 J/cm^2^ group. This decrease was a statistically significant compared to the control group. The results of MA relation to the use of three wavelengths (405 + 450 + 635 nm) are shown in Table 2 and Figure 7. The interaction between energy density and time of incubation after irradiation for mitochondrial activity is shown in Figure 2f.

After 24 h of irradiation with the 980 nm laser, an increase in MA activity was noted. At an energy density of 25 J/cm^2^, this increase was the highest (126.6%). However, it has not been shown that this increase was statistically significant compared to the control group. After 48 h of the exposure to all energy densities, a decrease in the MA value was noted (without statistically significant differences in relation to the control group). After 72 h, this decrease would deepen at all energy densities, reaching levels below 100% (without statistical differences compared to the control group). The results of MA relation to the use of the 980 nm laser irradiation are presented in Table 2 and Figure 8. The interaction between energy density and time of incubation after irradiation for mitochondrial activity is shown in Figure 2g.

After 24 h of the exposure, with the 1064 nm laser, the 3 J/cm^2^ energy density group showed a statistically significant decrease in relation to the control group. After the irradiation with an energy density of 25 and 64 J/cm^2^, there were no statistically significant differences in the MA value in relation to the control group. After 48 h, there was an increase in MA activity with 3 J/cm^2^ irradiation compared to the control group level. The groups with 25 and 64 J/cm^2^ did not differ statistically from the controls after 48 and 72 h. The highest MA value (104.0%) was obtained after 72 h at 64 J/cm^2^ energy density. The MA values for 1064 nm laser irradiation are shown in Table 2 and Figure 9. The interaction between energy density and time of incubation after irradiation for mitochondrial activity is shown in Figure 2g.

To sum up, the highest percentage of MA (122.1%) was observed in the group irradiated with the 635 nm laser, at an energy density of 64 J/cm^2^, after 48 h. The lowest percentage of MA (94.0%) was recorded in the group simultaneously irradiated with three wavelengths (405 + 460 + 635 nm). Noteworthy, the use of the 405 nm laser gave similar results to the 635 nm laser, although at a lower energy density of only 25 J/cm^2^. It is also important to observe that the standard deviation (SD) reached the highest value (SD = ± 71.7) in the case of 980 nm laser irradiation (after 24 h and at an energy density of 25 J/cm^2^) and 48 h after 635 nm laser irradiation at an energy density of 64 J/cm^2^ (SD = ± 34.5). In other cases, the SD did not exceed 11.1% (Table 3).

## 4. Discussion

As the main connective tissue cells, fibroblasts are responsible for the formation of the extracellular matrix and collagen. In the case of chronic stimuli leading to connective tissue damage, cell signaling and angiogenesis are disturbed and delayed [36].

When dealing with a deficit of the connective tissue, it seems important to supplement fibroblasts by implanting or, in the case of small defects, stimulating cells to migrate from the periphery of the wound. Yu et al. found a directly proportional relationship between the amount of fibroblast growth factor (FGF) and proliferating cells regenerating the gingiva–acellular cementum junction [37]. It has been demonstrated that the proliferation and migration of fibroblasts can be enhanced with the use of photobiomodulation, PBM. A wide spectrum of physical parameters that stimulate PBM, such as the light wavelength, power, exposure time, energy dose, frequency and type of laser, are administered [24,38]. However, there is no statement as to which wavelength, at what energy density, will result in the largest amount of fibroblasts. According to Sajan et al., it is the wavelength, not energy density, that affects oxidative stress and ATP levels, and thus also cellular proliferation [39]. Others think that energy density plays an important role.

Our experimental study was conducted in vitro. Fibroblasts were photobiomodulated in an optically transparent culture medium, in which cells should theoretically proliferate best at wavelengths of 400–600 nm. However, in vivo, cells should preferably proliferate at wavelengths of 600–1200 nm, due to the optical properties of the tissues [22,40,41,42]. This explains the use of so many different wavelengths. 

According to Chung et al., red/near infrared (NIR) light seems to be the most desirable for PBM effects due to its smaller dispersion. It is absorbed by tissue chromophores and therefore maximally penetrates the tissue, affecting cell proliferation and migration [43]. Ravera et al. observed the interaction of the 1064 nm Nd: YAG laser on mitochondrial aerobic metabolism and complex activity as well as ATP production [24]. 

The results of our pilot study, in the case of the group irradiated with the 635 nm laser, showed an increase in the mitochondrial activity in relation to the control group for all energy densities, after 24 and 48 h, which is of great importance in terms of the prevention of soft tissue scarring. Both the wavelengths of red and near infrared spectrum penetrate very well through the mucosa [16]. Therapy with these wavelengths decreases the level of IL-6 cytokine mRNA, which is responsible for scarring processes, so detrimental to the aesthetic effect [23,44,45]. Using a wavelength of 635 nm at 3 J/cm^2^ at a distance of 3 cm from the sample, Huang et al. found that by activating the Copper-glycyl-L-histidyl-L-lysin complex (Cu-GHK), red light increases the main secretion fibroblast growth factor, i.e., basic fibroblast growth factor (bFGF), and the production of procolagen 1 carboxyterminal propeptide (P1CP 9) which directly accelerates the wound healing process through its anti-inflammatory effects [46]. Cannarozzo et al. confirmed that 675 nm wavelength is useful to treatment of scars [47]. A statistically significant increase in fibroblast proliferation with red light was noted by most researchers at an energy density under 15 J/cm^2^. Sajan et al. found that the use of a wavelength of 636 nm at 25 J/cm^2^ reduces oxidative stress, which is important in modulating the proliferation and migration of fibroblasts and wound healing [39]. At the same time, both Sajan and Khan observed that the use of doses of over 15 J/cm^2^ in the case of the NIR laser has a phototoxic effect on cells. The increase in temperature activates, inter alia, heat shock protein 70 (HSP70)), generating harmful reactive oxygen species (ROS), damaging the functioning of the mitochondria [39,48]. Our study showed a statistically significant increase in the 64 J/cm^2^ group after 48 h. 

The advantage of low -power lasers over high-power lasers is that they cause lower thermal effects, which, combined with the lack of production of highly reactive oxygen species, do not damage cell membranes or DNA [49]. Our research shows that irradiation with a diode laser, with a wavelength of 980 nm for a power of 1–2 W, energy density of 3, 25 and 64 J/cm^2^ (PW) and duration of 3, 25 and 32 s does not have a significant effect on the biostimulation of fibroblasts, but it also does not cause cell apoptosis. Furthermore, an increase in the mitochondrial activity (MA) was observed after 24 h at all energy densities (3, 25 and 64 J/cm^2^). The highest MA increase was observed in the group treated with the energy density of 25 J/cm^2^. However, at 48 and 72 h, there was a decrease in cell proliferation in all three energy groups. This could be the result of increased expression of alpha-actin, a marker that differentiates fibroblasts into myofibroblasts [50,51]. A decrease in MA after 48 and 72 h was also observed by Kreisler et al. The authors emphasized the need for repeated irradiation of the treatment site every 24 h in clinical settings [52]. Similarly, in an in vivo study, Illescas-Montes et al. used a 940 nm diode laser at 0.5 W and 4 J/cm^2^ in two doses after 24 and 72 h. They observed a decrease in elastin expression, which has a positive effect on the regeneration of keratinized gingiva without leaving hypertrophic scars or keloids [53]. Our results correspond to the Engel’s report, stating that infrared spectrum exposure is more effective when using low power at high dose levels [54]. Gkogkos et al. proved that the use of high doses of energy at low power promotes fibroblast proliferation 48 h after irradiation, which in turn can induce the secretion of growth factors [55]. The use of the Nd:YAG laser with a power of 0.5–1.0 W at a density of 25 and 64 J/cm^2^ in impulse mode, at a distance of 5 mm from the cells did not cause fibroblast apoptosis but their proliferation, although the result was statistically insignificant. Most studies assessing cell proliferation and PBM effects in the infrared range involve diode lasers. Gkogkos et al., in the studies on the effectiveness of the Nd:YAG laser, observed a statistically significant increase in fibroblast proliferation in the group with the highest density used in the study, i.e., 15.8 J/cm^2^ using 0.5 W power after 48 h. The authors recommended the use of higher energy densities with less power [55]. Gutknecht et al. observed thermal cell damage and necrosis after Nd:YAG laser irradiation using a power of 2.1–3.0 W at a distance of 2 mm from the sample. They considered the exposure time to be 30 s or more as the reason [56]. Similarly, Almeida-Lopes et al. compared the efficacy of different wavelengths and they found that the infrared laser promoted gingival fibroblast proliferation with shorter cell exposure [57]. Our study did not confirm this theory. Sağlam et al. used Nd:YAG irradiation at a wavelength of 1064 nm 1.5–2.0 W, with exposure time of 10–20 s, at a distance of 2 mm from the sample. Neither cell apoptosis nor an increased release of growth factors was observed. A significant increase in the release of VEGF within 24 h was observed only in the group using the parameters of 2 W and the exposure time of 20 s [58].

Chen et al. noted the apoptosis of human fibroblasts in the gingiva induced by Nd:YAG laser radiation after the application of the power of 1.0–3.0 W and in direct contact with the cells. However, after irradiation of the cells from a distance of 2 mm, no significant decrease in fibroblasts viability was observed [59]. Therefore, comparing two types of light-red and near infrared, which should best stimulate fibroblasts to proliferate clinically, a statistically significant increase in MA in relation to the control group was recorded for the 635 nm laser.

Our research also assessed the stimulation of fibroblasts in the field of violet and blue light by means of a diode laser prototype constructed for the project. In this study, a statistically significant increase in MA after 48 h of irradiation with a 405 nm wavelength was observed for all energy densities of 3, 25 and 64 J/cm^2^. Importantly, it was comparable in each case. Among the categories of violet light, the range of approx 400 nm is the safest as it is not absorbed by any of the DNA bases. Simultaneously, wavelengths close to 400 nm can generate reactive oxygen species (ROS) at low doses, which are associated with the effects of PBM and do not damage DNA. Similarly, blue light has a stimulating effect on ROS production, although only in high doses [60,61,62].

It has an inhibitory effect on the proliferation, not only of cancer cells, but also of fibroblasts, as shown in our pilot study. There are reports in the literature of blue light stimulating fibroblast proliferation in vivo and in vitro. The observations included the direct contact of the laser tip with the keratinized gingival, due to the shallow penetration of this wavelength. In our study, the laser tip was at a 5.0 mm distance from the samples, as was the case with our other research groups. Therefore, it might be seen as the reason of the result of our study [63,64,65].

One of the aims of this pilot study was also to assess fibroblasts proliferation after using laser irradiation, with a beam composed of different wavelengths. Trelles et al. described in clinical studies the effects of a laser equipped with replaceable tips based on LEDs emitting light in the red and near infrared range. They used two wavelengths in the correct order: 830 ± 5 nm (near infrared) and 633 ± 3 nm (visible red), taking advantage of the fact that red light is focused on fibroblasts, and near infrared light is associated with the PBM of cells, which affects wound healing, and the transformation of fibroblasts into myofibroblasts. Both wavelengths increased the local blood supply. The combination was found to be more effective than using one of the wavelengths [66]. We did not find any research linking the same wavelengths as in our study in the literature. In our study, none of the combinations of different wavelengths in a pulse mode resulted in the expected statistically significant fibroblast proliferation. However, when combining the 405 and 635 nm wavelengths, no apoptosis was observed. On the other hand, proliferation similar to the control group was observed in the 64 J/cm^2^ group, after 72 h of incubation. The results were similar in the case of combining the 450 and 635 nm wavelengths, but proliferation was observed in the 25 J/cm^2^ group also after 72 h of incubation. In contrast, statistically significant cell apoptosis was observed in a group combining the 405, 450, and 635 nm wavelengths.

## 5. Conclusions

When using a 405 nm laser, a decrease in MA activity below 100% was observed on the first day after the exposure. A sudden, statistically significant increase in relation to the control group occurred after 48 h for energy densities of 3, 25, and 64 J/cm^2^. It is important that this increase was comparable for all energy densities.

In the case of the 635 nm and 980 nm lasers, an increase in mitochondrial activity was noted as early as on the first day after irradiation, but without statistically significant differences in relation to the control group.

The use of a 635 nm laser at the energy density of 64 J/cm^2^ resulted in the largest statistically significant increase in MA after 48 h in relation to the control group.

The initial decrease in the MA value after 24 h without statistical differences compared to the control, was observed in the groups irradiated with the 405, 450, 405 + 635, 450 + 635, and 1064 nm lasers. Statistically confirmed drop in MA after 24 h occurred when using the 405 nm laser (at 64 J/cm^2^ energy density), the 1064 nm laser (3 J/cm^2^ density) and the 405 + 450 + 635 nm lasers (for all energy densities).

The use of a 980 nm laser resulted in the highest MA value but without a statistical difference compared to the control group. Measurements of the MA activity in this group were burdened with the highest standard error.

The simultaneous use of three wavelengths (405 + 450 + 635 nm) caused a significant decrease in the MA value after 24 h.

Our study suggested that of all the lasers used the 635 nm and 405 nm prototype visible laser (developed at the Polish Academy of Sciences) which caused a statistically significant increase in the proliferation of gingival fibroblasts. Clinically, this laser can be useful in periodontological plastic surgery in terms of obtaining an optimal amount of fibroblasts for the regeneration of soft tissue. Deficiency of keratinized gingiva is a common clinical problem in case of orthodontic, prosthetic or implantology treatment, so soft tissue irradiation in band 635 and 405 nm may accelerate the healing process. 

## Figures and Tables

**Figure 1 jpm-11-00098-f001:**
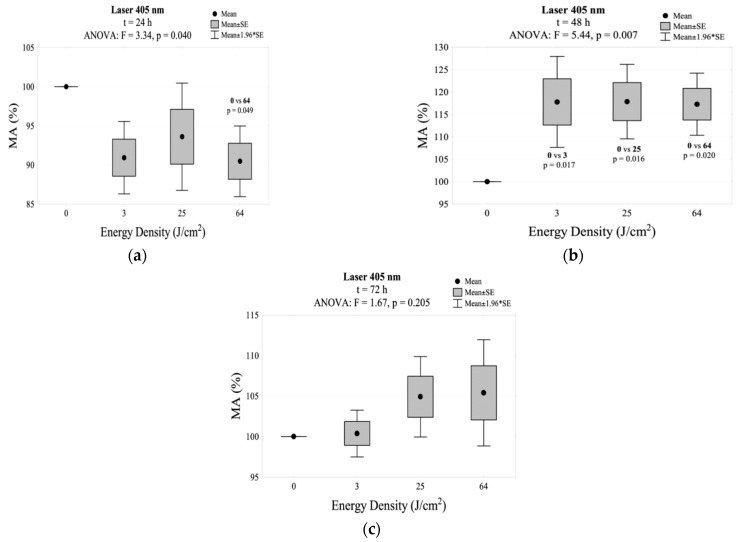
MA values measured (**a**) 24 h, (**b**) 48 h, and (**c**) 72 h after irradiation using the 405 nm laser.

**Figure 2 jpm-11-00098-f002:**
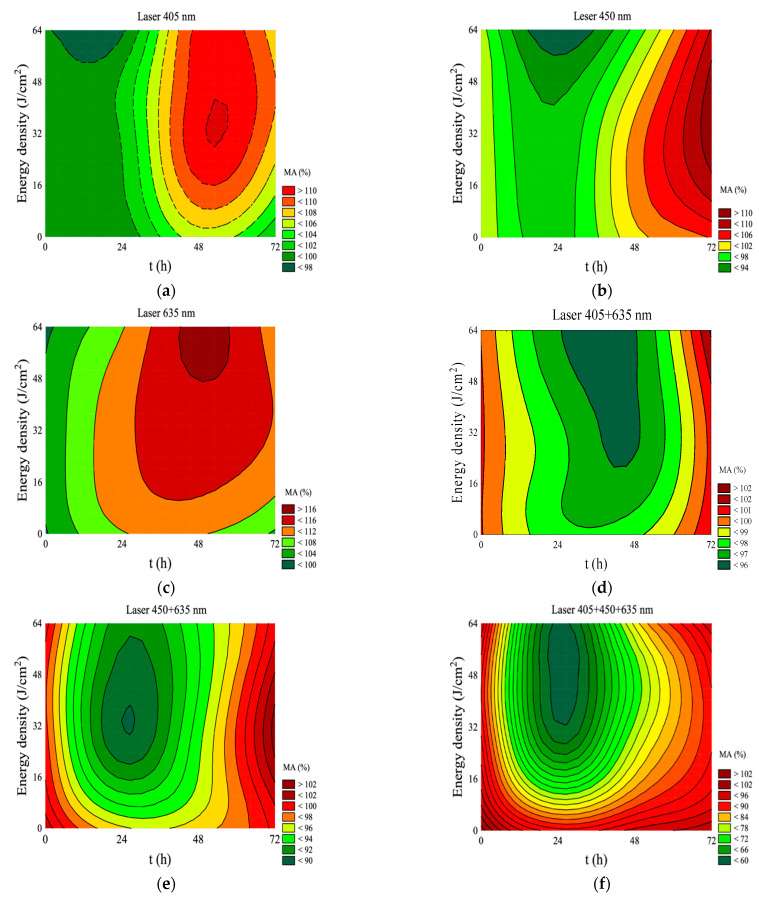
Interaction between energy density and time of incubation for mitochondrial activity after irradiation for: (**a**) 405 nm, (**b**) 450 nm, (**c**) 635 nm, (**d**) 405 + 635 nm, (**e**) 450 + 635 nm, (**f**) 405 + 450 + 635 nm, (**g**) 980 nm, (**h**) 1064 nm.

**Figure 3 jpm-11-00098-f003:**
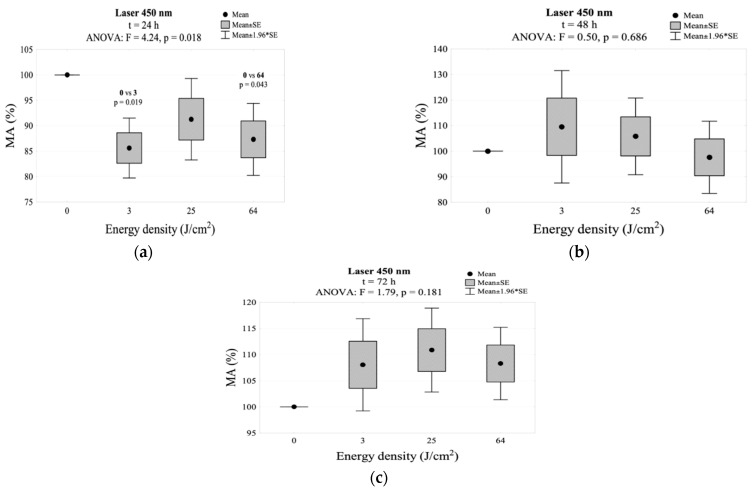
MA values measured (**a**) 24 h, (**b**) 48 h, and (**c**) 72 h after irradiation using the 450 nm laser.

**Figure 4 jpm-11-00098-f004:**
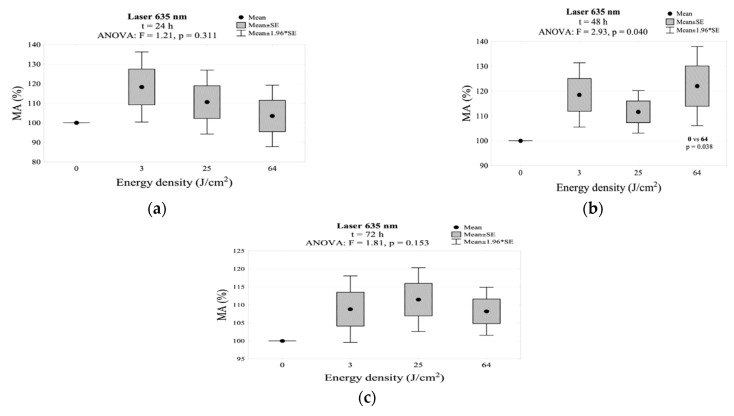
MA values measured (**a**) 24 h, (**b**) 48 h, and (**c**) 72 h after irradiation using the 635 nm laser.

**Figure 5 jpm-11-00098-f005:**
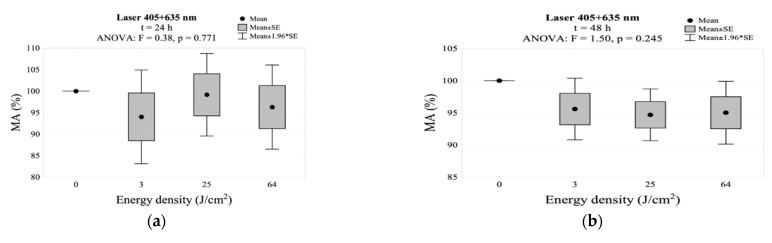
MA measured (**a**) 24 h, (**b**) 48 h, and (**c**) 72 h after irradiation using the 405 + 635 nm laser.

**Figure 6 jpm-11-00098-f006:**
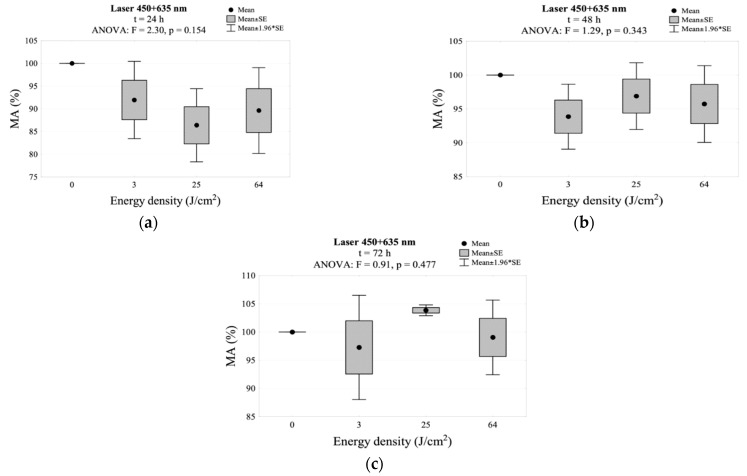
MA values measured: (**a**) 24 h, (**b**) 48 h, (**c**) 72 h. after irradiation using the 450 + 635 nm laser.

**Figure 7 jpm-11-00098-f007:**
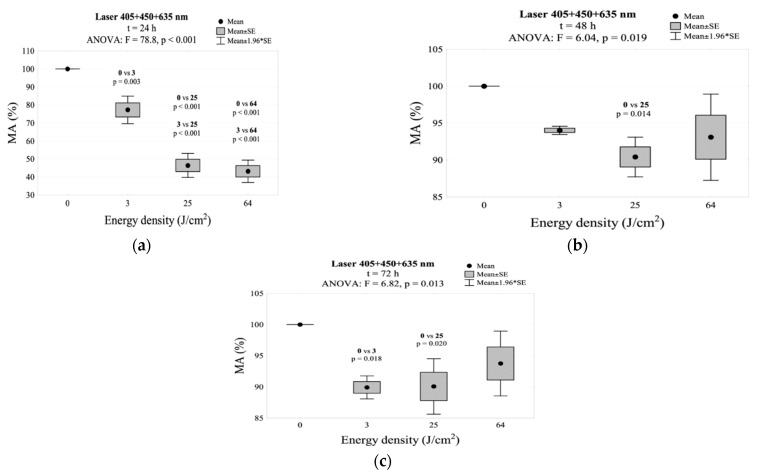
MA values measured: (**a**) 24 h, (**b**) 48 h, (**c**) 72 h after irradiation using the 405 + 450 + 635 nm laser.

**Figure 8 jpm-11-00098-f008:**
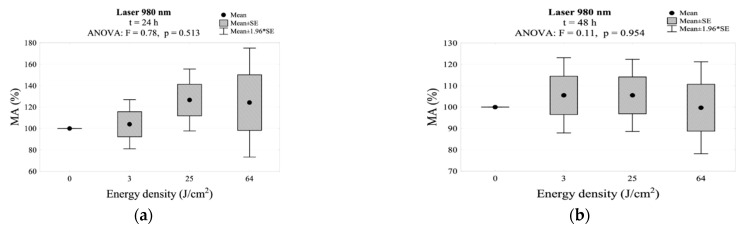
MA values measured: (**a**) 24 h, (**b**) 48 h, (**c**) 72 h after irradiation using the 980 nm laser.

**Figure 9 jpm-11-00098-f009:**
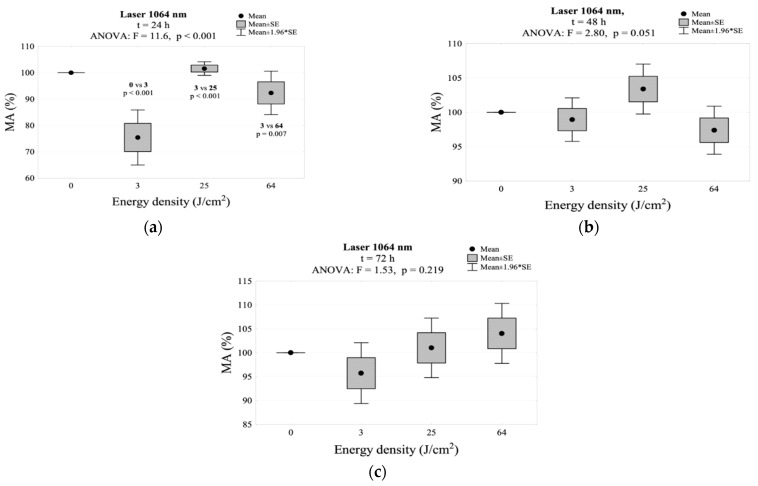
MA values measured: (**a**) 24 h, (**b**) 48 h, (**c**) 72 h after irradiation using the 1064 nm laser.

**Table 1 jpm-11-00098-t001:** Parameters of: Nd:YAG (LightWalker Fotona, Slovenia), 980 nm diode laser (Smart M, Lasotronix, Poland), and Visible laser developed for the project (Poland).

Type of Laser	Wavelength	Output Power for Energy Density			Pulse Modulation-Frequency
		3 J/cm^2^	25 J/cm^2^	64 J/cm^2^	
Nd-YAG (Light-Walker Fotona, Slovenia)	1064 nm	0.25 W	0.5 W	1.0 W	10 Hz
Diode laser (Smart M, Lasotronix Poland)	980 nm	0.50 W	1.0 W	2.0 W	
Visible diode laser developed for the project (Poland)	405 nm	0.25 W	0.5 W	0.7 W	10 Hz
	450 nm	0.25 W	0.5 W	1.0 W	
	635 nm	0.25 W	0.5 W	1.0 W	
	405 + 635 nm	0.15 W + 0.15 W	0.25 W + 0.25 W	0.5 W + 0.5 W	
	450 + 635 nm	0.15 W + 0.15 W	0.25 W + 0.25 W	0.5 W + 0.5 W	
	405 + 450 + 635 nm	0.10 W + 0.10 W + 0.10 W	0.15 W + 0.15 W + 0.15 W	0.35 W + 0.35 W + 0.35 W	

**Table 2 jpm-11-00098-t002:** Values of mitochondrial activity (MA) in function of wavelength, energy density (ED), and time (t) between irradiation and mitochondrial activity measurement (SE)—standard error, bold font- statistically significant difference versus control group, ref—control group).

		MA ± SE							
ED (J/cm)	t (h)	405 nm	450 nm	635 nm	405 + 635 nm	450 + 635 nm	405 + 450 + 635 nm	980 nm	1064 nm
0	0	100.0 (ref.)	100.0 (ref.)	100.0 (ref.)	100.0 (ref.)	100.0 (ref.)	100.0 (ref.)	100.0 (ref.)	100.0 (ref.)
0	24	100.0 (ref.)	100.0 (ref.)	100.0 (ref.)	100.0 (ref.)	100.0 (ref.)	100.0 (ref.)	100.0 (ref.)	100.0 (ref.)
0	48	100.0 (ref.)	100.0 (ref.)	100.0 (ref.)	100.0 (ref.)	100.0 (ref.)	100.0 (ref.)	100.0 (ref.)	100.0 (ref.)
0	72	100.0 (ref.)	100.0 (ref.)	100.0 (ref.)	100.0 (ref.)	100.0 (ref.)	100.0 (ref.)	100.0 (ref.)	100.0 (ref.)
3	0	100.0 (ref.)	100.0 (ref.)	100.0 (ref.)	100.0 (ref.)	100.0 (ref.)	100.0 (ref.)	100.0 (ref.)	100.0 (ref.)
3	24	90.9 ± 5.8	85.6 ± 7.4	118.4 ± 38.8	94.0 ± 13.6	91.9 ± 7.5	77.2 ± 6.8	104.0 ± 38.6	75.4 ± 18.5
3	48	117.8 ± 12.7	109.6 ± 27.5	118.5 ± 28.0	95.6 ± 6.0	93.9 ± 4.2	94.0 ± 0.5	105.5 ± 36.1	98.9 ± 5.6
3	72	100.4 ± 3.6	108.1 ± 11.0	108.8 ± 20.0	101.7 ± 10.0	97.3 ± 8.2	89.9 ± 1.6	87.1 ± 21.5	95.7 ± 11.2
25	0	100.0 (ref.)	100.0 (ref.)	100.0 (ref.)	100.0 (ref.)	100.0 (ref.)	100.0 (ref.)	100.0 (ref.)	100.0 (ref.)
25	24	93.6 ± 8.6	91.3 ± 10.0	110.6 ± 35.4	99.1 ± 12.0	86.4 ± 7.1	46.4 ± 5.9	126.6 ± 71.7	101.5 ± 4.6
25	48	**117.9 ± 10.4**	105.8 ± 18.8	111.7 ± 18.5	94.7 ± 5.0	96.9 ± 4.4	90.4 ± 2.4	105.5 ± 38.5	103.4 ± 6.4
25	72	104.9 ± 6.2	**110.9 ± 10.0**	111.5 ± 19.2	100.4 ± 21.4	**103.8 ± 0.8**	90.1 ± 3.9	87.6 ± 20.2	101.0 ± 11.0
64	0	100.0 (ref.)	100.0 (ref.)	100.0 (ref.)	100.0 (ref.)	100.0 (ref.)	100.0 (ref.)	100.0 (ref.)	100.0 (ref.)
64	24	90.5 ± 5.6	87.3 ± 8.9	103.5 ± 34.0	96.3 ± 12.2	89.6 ± 8.3	43.2 ± 5.5	124.4 ± 70.5	92.3 ± 14.6
64	48	117.3 ± 8.7	97.6 ± 17.6	**122.1 ± 34.5**	95.0 ± 6.1	95.7 ± 5.0	93.1 ± 5.2	99.7 ± 42.9	97.4 ± 6.2
64	72	105.4 ± 8.2	108.3 ± 8.6	108.2 ± 14.4	**102.5 ± 8.0**	99.0 ± 5.8	93.8 ± 4.6	84.9 ± 21.7	104.0 ± 11.1

**Table 3 jpm-11-00098-t003:** Maximum values of mitochondrial activity (MA^max^) for various wavelengths (λ), amounts of time after irradiation (t) and energy densities (ED) (SD-standard deviation).

MA^max^ (%)	Laser							
	A	B	C	D	E	F	G	H
λ (nm)	405	450	635	405 + 635	450 + 635	405 + 450 + 635	980	1064
ED (J/cm^2^)	25	25	64	64	25	3	25	64
t (h)	48	72	48	72	72	48	24	72
Mean ± SD	117.9 ± 10.4	110.9 ± 10.0	122.1 ± 34.5	102.5 ± 8.0	103.8 ± 0.8	94.0 ± 0.5	126.6 ± 71.7	104.0 ± 11.1

## Data Availability

The authors got acquainted with MDPI Research Data Policies. The data that support the findings of this study are available from the corresponding author upon reasonable request.

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
