# Peer review of "Assessment of Human Gingival Fibroblast Proliferation after Laser Stimulation In Vitro Using Different Laser Types and Wavelengths (1064, 980, 635, 450, and 405 nm)—Preliminary Report"

_jpm, 2021, doi:10.3390/jpm11020098_

Round 1

Reviewer 1 Report

Page 1, line 23: please replace thanks to with “due to”

Page 1, line 25: please explain the abbreviation MTT test

Page 1, line 38: the authors wrote about laser as a “diagnostic device”, even if afterwards it was mentioned as treatment device. Please explain.

Page 2, line 71: instead of biotope, I think the word “biotype” is more appropriate

Page 5, line 159: please correct “addition”

Please add during the conclusion or discussion part, the clinical relevance for readers (periodontists and general practitioners – dentist) and also, the authors should mention the importance of continuing with in vivo studies.

Author Response

Dear Reviewer,

We sincerely appreciate your reviewing our contribution and giving us another opportunity to improve our manuscript in the best possible way. Taking into account the Reviewers comments we have made a revision of our manuscript.

Reviewers’ response to the first revision:

  1. Page 1, line 23: please replace thanks to with “due to”

 The sentence was replaced from Thanks to with “Due to”.

  1. Page 1, line 25: please explain the abbreviation MTT test

The abbreviation MTT test was explained (Thiazolyl Blue Tetrazolium Blue).

  1. Page 1, line 38: the authors wrote about laser as a “diagnostic device”, even if afterwards it was mentioned as treatment device. Please explain.

Thank you for the significant remark. We added treatment device.

  1. Page 2, line 71: instead of biotope, I think the word “biotype” is more appropriate

We agree with you, thank you for the vigilance, the word was changed on “biotype”.

  1. Page 5, line 159: please correct “addition”

We corrected.

  1. Please add during the conclusion or discussion part, the clinical relevance for readers (periodontists and general practitioners – dentist) and also, the authors should mention the importance of continuing with in vivo studies.

We added your suggestions in conclusions section.

Best regards,

Authors.

Reviewer 2 Report

The article is a very interesting experimental study on the use of various lasers to assess human gingival fibroblast proliferation. The paper is clear and in my opinion eligible to publication after minor revision:

  Table 1 it is not clear to me why you didn't add also the results of 635 nm laser in this table. Please check

page 16 line 370-379 the results of the 635 nm are very interesting, as at this moment new lasers in the 600-700 nm range have been proposed for the treatment of collagen-based cutaneous diseases, such as scars or aging, histologically showing modification and rejuvenation of dermal collagen with very good clinical results; I think that this highlight would be a great add to your paper. Here I add some studies that I suggest you to read:doi: 10.1089/photob.2020.4908. doi: 10.1007/s10103-020-03063-6.

Author Response

Dear Reviewer,

We sincerely appreciate your reviewing our contribution and giving us another opportunity to improve our manuscript in the best possible way. Taking into account the Reviewers comments we have made a revision of our manuscript.

  1. Table 1 it is not clear to me why you didn't add also the results of 635 nm laser in this table. Please check

 Thank you for the vigilance, in the Table 1 - parameters of 635nm laser were added .

  1. page 16 line 370-379 the results of the 635 nm ….”

Thank you for the significant remark and suggestion. This knowledge certainly improve our manuscript in the best possible way. We added study (doi: 10.1007/s10103-020-03063-6.) to the references list.

Best regards,

Authors.
